EAD: effortless anomalies detection, a deep learning based approach for detecting outliers in English textual data

Wang Xiuzhe sciwxz@163.com
School of Foreign Languages, Zhengzhou College of Finance and Economics , Zhengzhou, Henan , China
Quadrini Michela
Electronic publication date: 2024 Nov 13
Publication date: 2024
Volume: 10
Electronic Location ID: e2479
Received 2024 Jun 28; Accepted 2024 Oct 14
Copyright: © 2024 Wang
Copyright year: 2024
Copyright holder: Wang
License: This is an open access article distributed under the terms of the Creative Commons Attribution License, which permits unrestricted use, distribution, reproduction and adaptation in any medium and for any purpose provided that it is properly attributed. For attribution, the original author(s), title, publication source (PeerJ Computer Science) and either DOI or URL of the article must be cited.
License URL: https://creativecommons.org/licenses/by/4.0/

Keywords: Outliers in textual data, Anomalies detection, LLM for anomalies

Funding: The authors received no funding for this work.

==============================
Anomalies are the existential abnormalities in data, the identification of which is known as anomaly detection. The absence of timely detection of anomalies may affect the key processes of decision-making, fraud detection, and automated classification. Most of the existing models of anomaly detection utilize the traditional way of tokenizing and are computationally costlier, mainly if the outliers are to be extracted from a large script. This research work intends to propose an unsupervised, all-MiniLM-L6-v2-based system for the detection of outliers. The method makes use of centroid embeddings to extract outliers in high-variety, large-volume data. To avoid mistakenly treating novelty as an outlier, the Minimum Covariance Determinant (MCD) based approach is followed to count the novelty of the input script. The proposed method is implemented in a Python project, App. for Anomalies Detection (AAD). The system is evaluated by two non-related datasets-the 20 newsgroups text dataset and the SMS spam collection dataset. The robust accuracy (94%) and F1 score (0.95) revealed that the proposed method could effectively trace anomalies in a comparatively large script. The process is applicable in extracting meanings from textual data, particularly in the domains of human resource management and security.

Introduction

Anomalies detection in the realm of data mining is the discovery of data points that deviate from the context of a document. The detection of anomalies is essentially used in different domains like business, human resource management and security, where the data comes from a simple textual script, customer review, or from a system log.

Being a classic problem in the domain of computer science (Mohaghegh & Abdurakhmanov, 2021), a number of algorithms have been proposed so far to gauge the deviation of data points from the baseline of a dataset (Chandola, Banerjee & Kumar, 2009). These purposely built models cover a number of areas, including fraudulent transactions (Garg et al., 2019), medical diagnosis (Zhang, Raghunathan & Jha, 2013), price management (Ramakrishnan et al., 2022) and network intrusion (Kwon et al., 2019).

The anomaly detection models can be categorized into supervised and unsupervised models. These models stumble when new anomalies are faced. In the case of supervised models, manual labeling is required, whereas with unsupervised models, outliers are detected based on specific attributes. Based on the encoding unit, the detection approach may either be character-based, word-based, or sentence-based. In character-level encoding (Mohaghegh & Abdurakhmanov, 2021; Rashid, Othman & Zainudin, 2019), outliers are detected based on the given character set. Such anomalies detection (AD) systems are suitable for short documents only. In the current era of artificial intelligence (AI), however, detection from larger datasets is needed. The word-level models are used for simple classification tasks like distinguishing spam and ham messages. For the word level outliers, typically, the bag-of-words (BOW) (Nowzohour et al., 2017) and the inverse-document-frequency (TF-IDF) (Spärck Jones, 2004) methods are used. Such systems fail to cope with the semantic meaning and context of a large corpus. The last category is sentence-level embedding, where neural network (N.N.) models are utilized (Bengio et al., 2003). Such systems are suitable for outlier detection based on the whole context of a script (Mikolov et al., 2013). Though the NN-based AD offers valuable insight, most of the system suffers from considering semantics (Arslan, Javaid & Awan, 2023). Moreover, the crucial flaws of such systems include extensive training and computational costs (Fan, Han & Liu, 2014). Hence, there is a need for an effortless method to deal with high-dimensional data with less computational cost.

This research work proposes an effective method for extracting anomalies in large-volume, high-variety data with promising accuracy. The model detects outliers in an unsupervised fashion-without labelled input and is implemented through a pipeline of two phases: the preparation phase and the application phase. Anomalies are determined based on contextual similarity with the centroid, as well as supporting novelty. In the preliminary phase, initially preprocessing is performed to remove noises from input data. Embedding is performed to have the latent space for computing centroid. The pre-trained model MinLM-l6-v2 (Galli, Donos & Calciolari, 2024) is exploited to have sentence-level embedding. Dimensionality reduction is performed to reduce the high-dimensional space to a lower dimension. In the application phase, anomalies are detected using the centroids of embeddings. To avoid mistakenly treating novelty as an outlier, the Minimum Covariance Determinant (MCD) based approach is followed to assess the novelty of the input script. The two versions of outliers, Near-anomalies and Real-anomalies, are detected for efficient analysis of input data. A schematic of the proposed method, which is implemented and evaluated in the Python project-App. forAnomalies Detection (AAD), is shown in Fig. 1. The interface of the project is designed in tkinter (Moore, 2021), with self-explanatory buttons to select datasets and find outliers. The outliers are displayed both in textual and graphical form. For the assessments, two independent datasets, the 20 newsgroups dataset from scikit-learn (Pedregosa et al., 2011) and the Kaggle English SMS spam collection dataset (Shafi’I et al., 2017), are used in the two separate evaluation sessions. In the first evaluation session, irrelevant texts were added incrementally to the 20 newsgroups dataset to check the outcomes of the model. In the second evaluation, the system was evaluated using messages in English taken from the Kaggle dataset. As a whole, a satisfactory accuracy score of 94% with an average F1 score of 0.95 was obtained. The outcomes of the analysis revealed the applicability of the proposed system in various domains. Moreover, the method is easily extendable to the realm of education and literature for text classification and coherent text generation.

Figure 1 Schematic of the proposed method.

The rest of the article is organized into four sections. In the literature review section, research works related with the proposed method are discussed. Details about the proposed system are covered in the methodology section. Evaluation and analysis are presented in the result section. The last section is the conclusion, which summarizes the entire research with future direction.

Literature review

Detection of anomalies is widely used in various domains like security, finance (Hilal, Gadsden & Yawney, 2022), and healthcare (Šabić et al., 2021). The aim of tracing the outliers in all such fields is to obtain meaningful insights (Esling & Agon, 2012) in textual data. Traditionally, methods based on statistical methods like exponential smoothing (Tang, Wang & Jiang, 2022) and ARIMA (Kozitsin, Katser & Lakontsev, 2021) were used. Such systems suffered flexibility besides handling different attributes in complex data. To overcome the issues, machine learning-based methods like auto-encoder (Thill et al., 2021) and isolation forests (Hongzuo et al., 2023) were introduced. With the advent of time, deep learning models like LSTM and CNNs (Schmidl, Wenig & Papenbrock, 2022) were utilized for promising results. However, these approaches also have their limitations, including high computational costs and the need for extensive training data (Schmidl, Wenig & Papenbrock, 2022).

For better understanding, researchers have categorized the literature of outlier detection into supervised, unsupervised and semi-supervised models (Boutalbi et al., 2023). With supervised methods, labeled data are to be fed to an ML-based model (Boutalbi et al., 2023). In such approaches, regression models are commonly utilized. The AD in jurisprudents is presented by Bobur et al. (2020). Similarly, for anomalous activities of hackers, a method based on logistics regression is given in Tharshini, Ragavinodini & Senthilkumar (2017). Although the accuracy of supervised-based systems is promising, such systems require a massive amount of manually labeled samples. Hence, the approaches often remained inappropriate in real-world business scenarios, particularly where the consistency of data needs to be checked in real-time. In unsupervised methods, similarities or dissimilarities are automatically traced without labeled inputs by machine learning (ML) classifiers. Most unsupervised models are utilized to find fake news on social media. The Unsupervised Fake News Detection Based on Autoencoder (UFNDA) system, as presented in Li et al. (2021), detects news in an unsupervised fashion.

Similarly, Eshraqi, Jalali & Moattar (2015) detect spam messages by auto-clustering short tweets. The unsupervised-based systems are typically used as a first attempt to understand the composition of data. With semi-supervised methods, not all but some input labeling is required for partial annotation. Such methods are typically suitable for unstructured data. The process of Steyn & de Waal (2016) is a semi-supervised approach that makes use of a multinomial naıve Bayes classifier for binary classification of anomalies in text documents.

With the emerging deep learning (DL) technology, cutting-edge models have been proposed for the effective detection of outliers. The DL-based method (Rettig et al., 2019) detects optimal energy consumption, and a model for outliers in intelligent grids is proposed in Liu & Nielsen (2016). Similarly, a framework based on principal statistical analysis is proposed in Xie & Chen (2017). Models based on the architecture of the transformer have also been suggested by Wang et al. (2022) and Zhang et al. (2021). As the pre-trained transformers model achieved commendable improvements in NLP-related tasks (Brown et al., 2005), the use of such models may enhance outlier detection, hence this research.

Materials and Methods

The proposed method intends to detect outliers using the unsupervised approach. The technique consists of two main phases- the preliminary phase and the application phase. After preprocessing in the first phase, embedding is performed using the pre-trained model, all-MinLM-l6-v2 (Galli, Donos & Calciolari, 2024). Dimensionality reduction of the sentence level embedding is performed to reduce the high-dimensional space to a lower dimension. To avoid mistakenly treating novelty as an outlier, the MCD-based approach is followed to assess the novelty of the input script. In the second phase, the centroids of embeddings are computed. Outliers are determined both on novelty and on the basis of the contextual similarity with the centroid. The two threshold levels are set as Near-anomalies and Real-anomalies, respectively, to detect possible and actual anomalies. Unlike other outlier detection systems (Schmidl, Wenig & Papenbrock, 2022; Boutalbi et al., 2023; Bobur et al., 2020; Tharshini, Ragavinodini & Senthilkumar, 2017), the method utilizes the cutting-edge all-MinLM-l6-v2 model for effective tracing of the anomalies in an unsupervised manner. Moreover, the two datasets containing text of different sizes and scripts are used to make the system applicable for general-purpose outlier detection. Details of each sub-process are given as follows.

Preliminary phase

The phase begins with loading the dataset and cleaning the data. Data was obtained from the cloud of Kaggle and scikit-learn, the 20 Newsgroups dataset (Moore, 2021) and the SMS spam collection (Shafi’I et al., 2017), respectively. Preprocessing is performed to remove extraneous words like emails, names and words with special characters. In order to obtain fast processing, text containing the first 4,000 words is taken from each class. The text column from the training set is converted to a data frame for effortless processing.

Sentence level embedding

Sentence embedding is the encoding of semantic information in a vector of real numbers. It is the numeric representation of text that can be used for semantic similarity and for the detection of outliers. The effective all-MiniLM-l6-v2 model is utilized to get the optimal sentence-level embedding. The all-MiniLM-L6-v2 is a pre-trained sentence embedding model that represents sentences in 384-dimensional dense vector space (Wang et al., 2020). It is comparatively a fast sentence embedding model with a satisfactory accuracy rate (Wilianto & Girsang, 2023). The architecture is much like the structure of the transformer models, i.e., BERT or RoBERTa. However, there are fewer layers and smaller hidden neurons in the model. There are a total of six transformer layers and 12 self-attention heads. With the self-attention heads, the relationship of each token in a sequence is captured. For effective training, the residual connections are used to connect each output component back to its original input. The sub-layer, Feed Forward Neural Network FNN, is applied to each token to learn complex relationships and make the model more robust. Lastly, layer normalization is performed to normalize activation and convergence. The architecture of the all-MiniLM-L6-v2 is given in Fig. 2.

Figure 2 The MiniLM-L6-v2 architecture.

Let S={s1,s2,..,sn} be the input text with ‘n’ sentences, the embedding E={e1,e2,…,en} is computed with the embedding function £ as, E = £(S). The pre-trained model effectively attains important semantic information contained in the text.

Dimensionality reduction

To transform the high dimensional space into lower dimension, the non-linear technique- t-Distributed Stochastic Neighbor Embedding (t-SNE) (Cieslak et al., 2020) and the Uniform Manifold Approximation and Projection (UMAP) (Ghojogh et al., 2021) are exploited. The t-SNE algorithm intends to map the high-dimensional space into the nearby points. The algorithm manages the divergence between the probability distributions of the pairwise similarities of the high-dimensional and lower-dimensional space. The cost function of t-SNE for the distribution of high-dimensional space (H) and lower-dimensional space (L) is given as;

(1) Cost(H,L)=∑i⁡∑jhijlog⁡hijlij

where hij and lij represent the conditional probabilities that points i and j are neighbors in the high-dimensional and lower-dimensional spaces, respectively. The pairwise similarities hij and lij are given as follows,

(2) hij=exp⁡(−(xi−xj)22σi2)∑k≠l⁡exp⁡(−(xk−xl)22σi2)

(3) lij=(1+((yi−yj)2)−1)∑k≠l⁡(1+((yk−yj)2))−1

where xi and xj are the input data points in the high-dimensional, yi and yj are the lower-dimensional embeddings and σ the Gaussian kernel, the small k and l are the indices representing individual data points in the dataset. UMAP is suitable for visualizing high dimensionality to lower while preserving the global structure. To get the lower dimensional data points, UMAP is utilized on the same dataset. A lower embedding dimension yu∈UMAPgenerated is selected if yu∈UMAPgenerated<yt∈UMAPt−SNE. The cross-entropy loss function of UMAP is given as,

(4) LCE=∑k<l⁡(Fslog⁡(FsPld)−(1−Fs)log(1−FsPld))

where Fs is the Fuzzy membership strength and Pld is the probability of that a data point has lower dimension.

Application phase

The application phase deals with actually tracing the outliers based on centroids of embeddings. Novelty in the text is also considered by using the MCD-based estimator. Once the input dataset is refined and reduced, the centroid of embedding and novelty are computed as discussed in the following subsections.

Centroid encoding

The centroid represents the crux context of a document (Brokos, Malakasiotis & Androutsopoulos, 2016). Sentences whose embeddings are closer to the centroid mean that the sentences are closely related to the context. In an n-dimensional embedding vector space, the closeness of sentences to the centroid is measured in terms of angles, as shown in Fig. 3.

Figure 3 Graphical representation of sentence embeddings, centroids, and dimension.

In the proposed method, centroids for the embedding vectors of the individual classes are computed. The centroid ‘C’ of an embedding vector E=(e1,e2,…en), is mathematically represented as;

(5) C=∑i=1n⁡ein.

If vj represents the embedding vector of jth class; vj=[xi1,xi2,…xim], then the Centroid List (CL) is computed as,

(6) CL=[∑xi1d,∑xi2d,.,.,∑ximd]

where ∑xi1 represents the sum of ith token of all vectors in a particular set vj and ‘d’ being the total number of vectors.

Novelty extraction

With a special module, novelty for each separate script is computed. The module begins by shaping the embedding into a matrix using the vstack method (Grus, 2019). Using the matrix as feature, the shape and location of central cloud data are computed. The MCD method (Hubert & Debruyne, 2010) is exploited for the purpose of using the MCD-based estimator, EllipticEnvelope (Ashrafuzzaman et al., 2020). The MCD estimator intends to discover such data points that minimize the determinants of their covariance matrix. Let H={x1,x2,…xn} be the set of ‘n’ data points in an m-dimensional space where xi∈Rm, The estimator (MCD) finds out Mahalanobis distance D(x) of x∈H as,

(7) D(x)=(x−CMCD)T∑MCD−1⁡(x−C)MCD

where CMCD is the centroid of the class.

The method fits the elliptic envelope to the central cloud. By using the envelope as a boundary, the normal and novel points are distinguished. The more embedding points lie outside the envelope, the more novelty there will be in the input text. The novelty score (NS) is normalized into the range [0,1]. Sentences less than or equal to 0.5 are considered novel, whereas greater than 0.5 is trivial.

Outlier detection

The designed Real and Near Outlier (R&N) algorithm, as shown in Algorithm 1, is followed to detect actual and near outliers. The function fR&N(E,CL,NS) returns outlier by taking Embedding E, Centroid list CL and Novelty score N.S. as input. The formula for real and near outlier is given as follows.

(8) Near_Outlier=Distancefromcentroid+2×Novelty

(9) Real_Outlier=Distancefromcentroid+3×Novelty.

Algorithm 1 Algorithm for computing real and near anomalies.

	

An embedding data point- ei∈E, is considered to be within the context if fR&N(ei,CL,NS)<NearOutlier.

Results

The proposed method is implemented in a case-study project-AAD in the VS-code with the Python libraries. For the implementation and evaluation of the project, a Corei7 laptop with 16 GB RAM, 4 GHz processor, 25 M.B. Cache, and RADEON graphics card was used. Details about the AAD project are given in the following subsection. The interface of ADS is designed in tkinter, which contains different buttons of self-explanatory captions, see Fig. 4. The AAD’s friendly interface contains three buttons: ‘Select DataSet’, ‘Find Anomalies’, and ‘Exit’. On the click event of the button ‘Select DataSet’, a file containing textual data is to be selected. Once a dataset is selected, outliers are found by clicking on the ‘Find Anomalies’ button. After preprocessing, embeddings, centroid and novelty are computed. Besides graphical output, outliers in the text are displayed on the left side of the window. The outliers detected in the four classes of the 20 Newspaper dataset are shown in Fig. 5.

Figure 4 Interface of the ADD project with the self-explanatory buttons.

Figure 5 Data points of the four classes with outliers (green circles) and centroids as ‘x’.

Performance of the proposed method was evaluated in two independent evaluation sessions. In the first evaluation session, irrelevant texts were added to the dataset incrementally to count occurrences of the outliers. For this purpose, the dataset was categorized into three sub-datasets: SD1–SD3. The sub-datasets by which the model was trained and the classes from which irrelevant text was borrowed are shown in Table 1.

Table 1 The training classes and the borrowed text classes.

Sub-dataset No.	Model trained class	Borrowed text class	
SD1	Comp.graphics	Politics.mideast	
SD2	Rec.motorcycles	Religion.misc	
SD3	Sport.hockey	Religion.christian	

For each SD, the system was assessed four times by adding different paragraphs with different numbers of words, as shown in Table 2.

Table 2 The number of borrowed text and detected outliers.

Sub-dataset No.	Assessment No.	No. of borrowed terms	Outliers	
SD1	1	97	71	
2	167	112	
3	213	177	
4	303	268	
SD2	1	45	34	
2	87	57	
3	137	99	
4	189	104	
SD3	1	63	52	
2	112	101	
3	176	145	
4	203	199	

It was observed that with the increase in the number of irrelevant texts, the extraction of outliers was increased accordingly. The linear relationship between the average detected outliers and average borrowed text for each of the SD is shown in Fig. 6. Similarly, more significant numbers of anomalies are detected if the value of the Near-outlier is low; see Fig. 7.

Figure 6 The graphs representing the linear relationship of borrowed text and number of outliers obtained in (A) SD1 (B) SD2 and (C) SD3.

Figure 7 The graph representing relationship between number of outliers with distance from the centroid.

Considering the irrelevant data points as anomalous, the F1 score was computed as,

(10) F1=2×Tp2×TP+FP+FN

where TP, FP, and FN represent True Positive, False Positive and False Negative, respectively. The satisfactory scores for sensitivity, specificity, precision and F1 for each of the S.D. are shown in Table 3.

Table 3 The analysis results of the first evaluation session.

Group	Sensitivity	Specificity	Precision	F1 score	
SD1	9.8	0.54	90.05	0.94	
SD2	0.94	0.55	0.908	0.92	
SD3	0.98	0.75	0.97	0.98	

As stated in Chicco & Jurman (2020), the Matthews correlation coefficient (MCC) is more reliable than the F1 score; for further assessment, the MCC score was computed using the given equation;

(11) MCC=(TP×TN)−(FP×FN)(TP+FP)×(TP+FN)×(TN+FP)×(TN+FN)(TP×TN)−(FP×FN)

A satisfactory MCC score of 0.6 was obtained, showing better performance of the model.

In the second evaluation session, the system was evaluated by SMS messages from the kaggle dataset (Shafi’I et al., 2017). Though the dataset has 5,574 English messages being tagged as ham (legitimate) or spam, 4K random SMS were selected. It was observed that 94% of the spam messages were correctly traced as outliers, see Table 4. To compare accuracy of the proposed method, relative analysis was conducted with the state-of-the-art systems, see Table 5. In the table, F1 score against the proposed model is mean of the individual F1 scores obtained for the subdatasets; SDi|i=13.

Table 4 Analysis results of the second evaluation session.

Dataset	Sensitivity	Specificity	Precision	Accuracy	F1 score	
The Kaggle SMS spam collection	0.98	0.97	0.987	0.94	0.98	

Table 5 Comparison of the state-of-the-art methods on the basis of F1 score.

Author(s)	F1 score	Remarks	
Su et al. (2019)	0.86	Stochastic RNN based	
Faber, Pietron & Zurek (2021)	0.67	Neuro-evolution methods are utilized	
Krajsic & Franczyk (2021)	0.88	Deep generative model is used	
Sahu & Mukherjee (2020)	0.73	Based on the Gaussian Mixture Model (GMM)	
Zhang et al. (2021)	0.89	Variational Auto-encoder (VAE) is exploited	
Xiuzhe Wang (the proposed system)	0.95	All-MiniLM-L6-v2 based	

Conclusions

Detection of text anomalies plays a significant role in various domains for estimating customer satisfaction, product quality, anticipating security risks and other rare events. Normally, the outliers exhibit variations in customer likes-dislikes, security risk, or ambiguous financial transactions. Most of the proposed systems are either data-hungry or computationally costlier. With this work, a sentence-level anomalies detection method is introduced based on the cutting-edge model of all-MiniLM-L6-v2. The method makes use of embedding centroids to extract outliers in the textual data. Unlike other models, the MCD-based approach is followed to support novelty. Hence, novel sentences in the context are not treated as outliers. The method is evaluated in two separate sessions by two non-related datasets with mixed-type contents. For better assessment, non-related texts were added incrementally during the evaluation. Satisfactory accuracy, precision and F1 scores (0.95) were obtained, showing the applicability of the method in a wide range of applications, particularly in text classification and summarization. As no separate tools are used for auto-encoding, tokenization, and pooling, the method ensures less computational cost. Besides the detection of outliers, the model is suitable for the analysis of textual data in business and academic domains.

Moreover, the method is easily extendable to the realm of education and literature for text classification and coherent text generation. It was observed that an increase in the number of sentences in individual paragraphs depleted the performance of the method. In our future work, we aim to enhance the system to avoid the said limitation and to support an even larger dataset of the Reuters Corpus Volume I (RCV1)-the archive of over 800,000 stories.

Supplemental Information

Supplemental Information 1 Code Interface.

Supplemental Information 2 Description of AI Application.

Supplemental Information 3 Dataset.

Supplemental Information 4 Code.

Additional Information and Declarations

Competing Interests

Author Contributions

Data Availability

The authors declare that they have no competing interests.

Xiuzhe Wang conceived and designed the experiments, performed the experiments, analyzed the data, performed the computation work, prepared figures and/or tables, authored or reviewed drafts of the article, and approved the final draft.

The following information was supplied regarding data availability:

The 20 Newsgroups dataset and code are available at Kaggle:

https://www.kaggle.com/datasets/crawford/20-newsgroups.

The SMS Spam Collection dataset and code are available at Kaggle:

https://www.kaggle.com/datasets/uciml/sms-spam-collection-dataset.

Data is available at Zenodo:

Xiuzhe, W. (2024). EAD: Effortless Anomalies Detection, A deep learning based approach for detecting outliers in textual data [Data set]. Zenodo. https://doi.org/10.5281/zenodo.13847878.

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
