# Peer review of "EAD: effortless anomalies detection, a deep learning based approach for detecting outliers in English textual data"

_PeerJ Computer Science, doi:10.7717/peerj-cs.2479_

## Round 0.1 · original submission · Major Revisions

Your manuscript has been assessed by our reviewers. They have raised a number of points which we believe would improve the manuscript and may allow a revised version to be published in PeerJ Computer Science.

Reviewer 1 ·

Basic reporting

> Clear and unambiguous, professional English used throughout.

Yes

> Intro & background to show context. Literature well referenced & relevant.

The introduction is full of references but there are briefly described.

> Structure conforms to PeerJ standards, discipline norms, or any deviations are to improve clarity.

Yes

> Does the Introduction adequately introduce the subject and make it clear what the motivation is?

Partially

> Formal results should include clear definitions of all terms and theorems, and detailed proofs (where necessary).

Yes but there are mistakes in the formulas.

Experimental design

> Article content is within Aims and Scope of the journal and article type.

Yes

> Rigorous investigation performed to a high technical & ethical standard.

Yes

> Methods described with sufficient detail & information to replicate (code, dataset, computing infrastructure, reproduction script, etc.).

The method is described carefully but the software code is unavailable.

> Is there a discussion on data preprocessing and is it sufficient/required?

Yes

> Are the evaluation methods, assessment metrics, and model selection methods adequately described?

Yes

> Are sources adequately cited? Quoted or paraphrased as appropriate?

Yes

Validity of the findings

> Impact and novelty is not assessed. Meaningful replication is encouraged where rationale & benefit to the field is clearly stated.

Partially

> Conclusions are well stated & limited to supporting results.

Partially: conclusions are too short.

> Are the experiments and evaluations performed satisfactorily?

Yes

> Is there a well-developed and supported argument that meets the goals set out in the Introduction?

Yes

> Does the Conclusion identify unresolved questions / limitations/ future directions?

No

Additional comments

Major points
1- The formulas (1) and (2) at page 9 are unreadable
2- Results are measured only through accuracy and F1 score, which are misleading metrics ( https://doi.org/10.1186/s12864-019-6413-7 ). The results should be reported and discussed as Matthews correlation coefficient, sensitivity, specificity, precision, and negative predictive value, too.
3- It is unclear why the author decided to use t-SNE rather than using UMAP. Results should be regenerated through UMAP.

Minor points
4- There is a number of format mistakes that should be fixed:
"python" --> "Python"
"ADD" --> "AAD"
"scikit-learn-" --> "scikit-learn"

Reviewer 2 ·

Basic reporting

The author proposed a pipeline that remove the anomalies from textual data in unsupervised. The pipeline consist of two phases. The preprocessing phase and application phase. In preliminary phase the author removes the extraneous words such as emails, names. Then use MiniLM-16-v2 for word embedding while in applications phase the author used centroid base techniques to check that how a sentence is related to its centroid, and then check the novelty using Minimum Covariance Determinant (MCD).

Suggestions:

1:- The paper is well written, however, it need to optimized the introduction section by including one paragraph to discuss organization of the paper.
2:- The author also include one paragraph that differentiate his work from the closely related work in term of methodology, datasets or models used for experiments.
3:- The author used MiniLM-16-v2 model for embedding and cite [14] paper in introduction section while in Material & Method section cite [13] paper create confusion. The author should take care of such citation carefully and cite reference [14].
4:- In [14] the author named the "all-MiniLM-16-v2" while the paper mentioned it MiniLM-16-v2. is this new model? or used the same model as in [14].
5:- The author used UFNDA acronyms for the first time and didn't define it. It should be define for the first time and then used the acronym.

Experimental design

1:- The author did not compare the results with other well know approaches. If the this baseline than should clearly mentioned in abstract or introduction.
2:- It would be better and will strengthen the results if it is compare with reference such as [36].
3:- The author should include a separate section for the first paragraph in Results section that describe the interface of the developed application.
4:- The paper include just figure for the algorithm of real and near anomalies should be cite if it is taken from the other work. It would be better if the author write the steps in latex or world file.
5:- How the author choose the Real (0.8) and Near (0.5) anomalies thresholds ?
6:- The captions for Fig 1 and 2 should be revised.

Validity of the findings

The author proposed pipeline that find the anomalies in sentence using deep learning approach and received accuracy 94% and F1 score 0.955 mention in abstract while in Table while in Table 4 it is 94% accuracy and 0.98 F1 score. 0.955 is not mention in Table 3 as well as Table 4. The author should clarify the results.

Additional comments

The Figures, Tables goes to separate pages at the end of the paper make it difficult for me to check the figure and results. It would be better that each figure and Table should be place at specified position. Moreover, all the equations are not in a proper format.

---

## Round 0.2 · Minor Revisions

Please, take into account the additional final comments of Reviewer 2.

Reviewer 1 ·

Basic reporting

The authors addressed my comments correctly and therefore I can recommend this article for acceptance.

Experimental design

Methods described with sufficient detail & information to replicate (code, dataset, computing infrastructure, reproduction script, etc.).

Yes

Is there a discussion on data preprocessing and is it sufficient/required?

Yes

Are the evaluation methods, assessment metrics, and model selection methods adequately described?

Yes

Validity of the findings

Impact and novelty is not assessed. Meaningful replication is encouraged where rationale & benefit to the field is clearly stated.

Yes

Are the experiments and evaluations performed satisfactorily?

Yes

Does the Conclusion identify unresolved questions / limitations/ future directions?

Yes

Reviewer 2 ·

Basic reporting

The author incorporated the suggested changes and it is ok for me now.

Experimental design

ok

Validity of the findings

I Table 5, Proposed system produce 0.95 F1 score. If it is the mean of F1 score for sub datasets SD1, SD2 and SD3 than mention a single sentence about it where you cite Table 5.

Additional comments

1:- In last paragraph of introduction section... The author remove either literature review or related work because both refer the same thing.
2:- In methods and material section.... The first sentence should be revised because unsupervised and data without label refer the same meaning. It may be like this. "The proposed method intends to detect outliers using unsupervised method" Or the author is free to rephrase it.
3:- make sure you have write the "scikit-learn" in your draft.
4:- The author used Table 5 for comparison with other approaches. Make sure that you run your test set on these approaches or The authors have test their model on your test data.
5:- In table 5 F1 score is 0.95 ...I think it is the mean of F1 score for SD1, SD2 and SD3 dataset. If yes than write a single sentence about it in conclusion or where you cite Table 5.

---

## Round 0.3 · accepted · Accept

The authors have addressed all of the reviewers' comments and the manuscript is ready for publication.